# Effects of Biomaterials Derived from Germinated Hemp Seeds on Stressed Hair Stem Cells and Immune Cells

**DOI:** 10.3390/ijms25147823

**Published:** 2024-07-17

**Authors:** Donghyun Kim, Namsoo Peter Kim, Boyong Kim

**Affiliations:** 1Department of Food Science and Biotechnology, Andong National University, Andong 36729, Republic of Korea; kimdhgo@student.anu.ac.kr; 2Center for Cannabis Certificate, Makers’ Station, Washington, DC 20001, USA; nsking21@gmail.com; 3Department of Pharmacognosy and Pharmaceutical Botany, Chulalongkorn University, Bangkok 10330, Thailand; 4EVERBIO, 131, Jukhyeon-gil, Gwanghyewon-myeon, Jincheon-gun 27809, Republic of Korea

**Keywords:** hemp, callus, alopecia, hair stem cell, dihydrotestosterone

## Abstract

Androgenetic alopecia is a genetic disorder that commonly causes progressive hair loss in men, leading to diminished self-esteem. Although cannabinoids extracted from *Cannabis sativa* are used in hair loss treatments, no study has evaluated the effects of germinated hemp seed extract (GHSE) and exosomes derived from the calli of germinated hemp seeds on alopecia. Therefore, this study aimed to demonstrate their preventive effects against alopecia using various methodologies, including quantitative PCR, flow cytometry, ELISA, and immunocytochemistry. Our research highlights the preventive functions of GHSE (GE2000: 2000 µg/mL) and exosomes from the calli of germinated hemp seeds (E40: 40 μg/mL) in three biochemical categories: genetic modulation in hair follicle dermal papilla stem cells (HFDPSCs), cellular differentiation, and immune system modulation. Upon exposure to dihydrotestosterone (DT), both biomaterials upregulated genes preventing alopecia (*Wnt*, *β-catenin*, and *TCF*) in HFDPSCs and suppressed genes activating alopecia (*STAT1*, *5α-reductase type 1*, *IL-15R*). Additionally, they suppressed alopecia-related genes (*NKG2DL*, IL2-Rβ, JAK1, STAT1) in CD8^+^ T cells. Notably, E40 exhibited more pronounced effects compared to GE2000. Consequently, both E40 and GE2000 effectively mitigated DT-induced stress, activating mechanisms promoting hair formation. Given the limited research on alopecia using these materials, their pharmaceutical development promises significant economic and health benefits.

## 1. Introduction

Alopecia is the phenomenon of hair loss from one’s head or other parts of the body due to various causes, including genetics, stress, and diseases [1]. This condition results in psychological effects such as low self-esteem in patients [1]. Androgenetic alopecia (AGA) is the most common type of alopecia and causes the most progressive hair loss [2]. Generally, the hair growth cycle comprises four phases: anagen, catagen, telogen, and exogen [3]. The cycling of these phases is dependent on the condition of the follicles that produce hair [1]. The most important phase for hair growth is the anagen phase; approximately 90% of hair growth in a healthy scalp occurs in this phase [4]. However, under a short anagen phase, the hairs progressively become weaker and resemble vellus hairs [1]. 

Androgen effects on the human skin include the growth and differentiation of sebaceous glands, wound healing, hair growth, and the regulation of epidermal barrier functions [5]. Although thyroid hormones and glucocorticoids are involved in hair growth [6], androgens are the key hormones that regulate the inhibition of terminal hair growth [2]. Paradoxically, although androgen-dependent areas are large, the hormone suppresses hair growth and promotes a short anagen stage in scalp follicles [2]. Hair follicle dermal papilla cells (HFDPCs) play various roles, including the maintenance of epithelial cell growth, the mediation of androgenic signals by paracrine growth factors, and productive modulation of factors from the epithelial cells of hair follicles [7,8,9]. Additionally, androgen action in hair follicles depends on local bioavailability. Therefore, even if androgens are at a normal level in the circulatory system, both testosterone and dihydrotestosterone (DT) are increased in local regions in androgenic alopecia. 

Hemp, *Cannabis sativa*, is cultivated for industrial purposes, including uses in food, paper, rope, textiles, clothing, bio-plastics, biofuel, and animal feed [10,11]. Cannabinoids are the most notable molecules among the bioactive compounds in hemp. Although at least 113 phytocannabinoids in hemp have been isolated, the 6 key cannabinoids are tetrahydrocannabinol (THC), cannabidiol (CBD), cannabigerol (CBG), cannabinol (CBN), cannabichromene (CBC), and tetrahydrocannabivarin (THCV) [12]. CBD, THCV, and cannabidivarin (CBDV) are involved in hair regrowth therapies for alopecia [13], while hemp seed oil improves hair growth [14]. However, there is no research on the effects of germinated hemp seed extract (GHSE) and exosomes derived from the callus of the germinated seeds on alopecia. 

The callus is a mass of parenchymal cells and is induced from explants using a culture medium supplemented with various hormones including auxin, cytokinin, and gibberellin [15]. The ratio between 6-benzylaminopurine (6-BAP, cytokinin) and 1-naphthaleneacetic acid (NAA, auxin family) is crucial for the induction and growth of hemp calli [16]. Recent research suggests that callus-derived exosomes impact various bioactive functions including the activation of osteogenic differentiation and immunomodulation [17]. However, there is no research on the bioactive function of callus-derived exosomes from germinated hemp seeds. 

Exosomes, approximately 40–100 nm in size, are secreted by almost all types of cells in organisms [18,19]. However, exposure to specific conditions can induce cells to secrete exosomes containing diversely altered components such as proteins, carbohydrates, mRNA, microRNA (miRNA), and DNA molecules [20]. The surrounding cells receive various stimulations from these induced exosomes via their altered components. Owing to these characteristics, exosomes have the potential to be applied as biomaterials in various fields, including foods, pharmaceuticals, and cosmetics [21]. 

Based on a Japanese report [22], the total nationwide cost attributed to alopecia was estimated to be JPY 112.7 billion (approximately USD 857 million), with productivity loss accounting for JPY 88.1 billion (78.2%). Approximately over 2 million days of activity time are lost annually due to alopecia. These economic and time burdens adversely affect people. Additionally, treatments for alopecia deteriorate the quality of life due to side effects including ejaculation disorders, breast tenderness, testicular pain enlargement, dermatitis itching, flushing, and headaches [23]. Therefore, it is necessary to develop biological materials with minimal side effects and excellent efficacy to address these issues in alopecia treatment.

Therefore, this study aimed to demonstrate the functions of GHSE and exosomes derived from the calli of germinated hemp seeds for the prevention of alopecia.

## 2. Results

We documented the biological functions of two biomaterials, GHSE and exosomes derived from the calli of germinated hemp seeds, including the activation of cellular differentiation, the modulation of alopecia stimulation between HFDPCs and immune cells, and the activation of immune cells against pathogens despite exposure to DT.

### 2.1. Suppression of Alopecia Markers in HFDPSCs by Two Materials

Alcoholic extracts (AEs) 50 (extraction of the germinated hemp seeds with 50% ethanol) were the most effective among various extracts (Figure 1a). Based on the results for cytotoxicity in HFDPSCs, the established exposure doses of the two materials, AE50 and the induced exosome derived from the calli of germinated hemp seeds, were 2000 µg/mL and 40 μg/mL, respectively (Figure 1b–d). The dose of DT was established to be 1 ng/mL for HFDPSCs (Figure 1e,f). For alopecia-modulating markers, although exposed to DT1, the two materials (GE2000 and E40) upregulated alopecia by preventing the activation of genes for proteins including TCF (T cell factor), β-catenin, and Wnt (Wingless and Int-1) in HFDPSCs. Notably, E40 dramatically upregulated genes in HFDPSCs (Figure 2). Contrary to these results, alopecia-activating genes were downregulated by GE2000 and E40 in HFDPSCs (Figure 2). Compared with DT, TCF, Wnt, IL-15R (Interleukin-15 receptor), and NKG2DL (natural killer group 2, member D ligand), genes were dramatically modulated by E40 in HFDPSCs (Figure 2). 

### 2.2. Activation and Protection of Differentiation in HFDPSCs by Two Materials

In contrast to DT1, GE2000 and E40 activated and protected the differentiation of HFDPSCs into HFDPCs and germ cells (Figure 3a,b). Compared with GE2000, E40 activated and protected the differentiation more strongly. The effects of E40 were approximately 1.6 times higher than those of GE2000 (Figure 3b). Notably, E40 activated HFDPSC differentiation into germ cells more intensely than into HFDPCs (Figure 3a,b). The flow cytometry results from the evaluation of E40 and GE2000 induction (Figure 3b) corresponded with their immunocytochemistry results (Figure 4). 

### 2.3. Prevention of Alopecia by GE2000 and E40 in Immune Cells

In CD8^+^ T cells, GE2000 and E40 downregulated alopecia-activating markers including IL2-Rβ, JAK1 (Janus kinase 1), and STAT1 (signal transducer and activator of transcription 1) under DT1 influence (Figure 5). Compared with the other two genes, STAT1 protein expression was dramatically downregulated by activated HFDPSCs with the two materials despite exposure to DT1 in CD8^+^ T cells (Figure 5). On average, the intensities for cells expressing STAT1 were 3.8 times higher than those expressing DTCM (Figure 5).

Under the various conditioned media, CD8^+^ T cells modulated their secretion of IFNγ (interferon gamma) (Figure 6). Notably, when exposed to E40CM, T cells suppressed the secretion of IFNγ compared to those under DT1. The secreted concentrations of IFNγ under E40CM were approximately 8.2 times lower than those under DTCM influence (Figure 6c).

### 2.4. Comparison of Effects of Hemp Seed Oil and Two Materials

Compared with HSO, GE2000 and E40 upregulated the alopecia-preventing genes more intensely, including the genes for TCF, β-catenin, and Wnt in HFDPSCs, despite exposure to DT1 (Figure 7). Although the preventive effects of GE2000 were slightly higher than those of HSO, the effects under E40 were approximately 2.2 times higher than those of hemp seed oil (HSO) (Figure 7). Moreover, HSO is effective in preventing alopecia based on recent studies [24,25]. However, this study suggests that there are more effective biomaterials against the alopecia-inducing compound DT (Figure 7).

## 3. Discussion

Although hemp utilization has been restricted by cannabis regulations in multiple countries, recently, several nations have eased or abolished these restrictions to foster the development of the hemp industry [26]. Moreover, industrial hemp has been cultivated and exported by more than 30 countries, contributing to the development of various products [27]. 

Although not life-threatening, androgenic alopecia has a negative effect on the patient’s quality of life, mental health, and work productivity [28]. Therefore, the development of materials with effective and eco-friendly results without side effects is a necessity. Some studies [29,30,31] have reported that seed oil and CBD in hemp are associated with hair regrowth. However, their use has disadvantages, including difficulties in application, usage limitations, and side effects, which limit their impact on preventing androgenic alopecia [31,32]. Further, there are no reports of hemp-derived materials associated with the activation of follicle cell differentiation from HFDPSCs and protection against DT. 

This study reveals the preventive functions of two biomaterials (GE2000 and E40) for androgenic alopecia through three biochemical categories, including genetic modulation in HFDPSCs, cellular differentiation, and the modulation of the immune system.

First, in the genetic modulation of HFDPSCs, two materials modulated alopecia-associated genes, including genes for TCF, β-catenin, Wnt, STAT1, 5α-reductase type 1, IL-15R, and NKG2DL. Notably, E40 dramatically upregulated TCF, β-catenin, and Wnt levels in HFDPSCs despite exposure to DT (Figure 2). Additionally, both GE2000 and E40 intensely downregulated alopecia-activating genes, including genes for IL-15R and NKG2DL (Figure 2). Recent studies have shown that the Wnt signaling pathway is a crucial key to hair growth [33,34]. Various types of Wnt molecules (Wnt1a, Wnt3a, Wnt4, Wnt5a, Wnt7b, and Wnt10a/b) play important roles, including hair follicle formation, damage repair, anagen gene expression, regeneration, and wound healing, in the hair growth cycle [33,34]. Further, cascadic signals associated with Wnt, the activation of β-catenin, and TCF activate hair growth [33,34]. Additionally, expressive suppression of STAT1, 5α-reductase type 1, IL-15R, and NKG2DL molecules in follicular epithelial cells is crucial to prevent alopecia [31]. Conversely, the downregulation of 5α-reductase type 1 in hair follicular epithelial cells suggests suppression of androgenic alopecia [32]. Notably, the expression of STAT1, IL-15R, and NKG2DL in epithelial cells activates alopecia through communication with CD8^+^ T cells [31]. These reports suggest that the two biomaterials prevent androgenic alopecia through the modulation of alopecia-associated genes in HFDPSCs. Additionally, the significant effect of E40 indicates the possibility for pharmaceutical materials to prevent androgenic alopecia. 

Second, based on the results for cellular differentiation, the two materials accelerated germ cell and HFDPC differentiation from HFDPSCs despite exposure to DT. As seen in Figure 3 and Figure 4, GE2000 and E40 activated germ cell and HFDPC differentiation from HFDPSC. Notably, differentiation into germ cells was intensely activated. Ordinarily, Itg9 and Sox2 molecules are marker proteins for HFDPCs, and P-cadherin and K15 are also markers for hair germ cells [35]. A hair bulb located at the lowest part of the hair follicle, where the hair germ is positioned, is a trigger for hair regeneration. The hair germ cells communicate with HFDPCs, induce stem cell activation in the hair bulge, and initiate the anagen phase [36]. Therefore, GE2000 and E40 are worth exploring as key biomaterials for alopecia-preventing items in the cosmetic and pharmaceutical markets. 

Third, alopecia-activating molecules in CD8^+^ T cells were modulated by HFDPSCs activated by the two materials. The signal pathways associated with IL2-Rβ, JAK1, and STAT1 in CD8^+^ T cells are involved in alopecia [31]. Moreover, IFNγ and various other cytokines are pivotal players in alopecia. IFNγ and γ_c_ cytokines (IL-2, IL-7, and IL-15) are involved in the activation of Janus kinases (JAKs) and signal transducers and activators of transcription (STATs) [37]. Thus, the inhibition of this signaling pathway is a crucial mechanism of alopecia prevention and therapy. As seen in Figure 5 and Figure 6, the two materials suppressed the expression of these molecules in CD8^+^ T cells despite exposure to DT. Notably, E40 suppressed them more strongly compared to GE2000. These results suggest that E40 can be a possible bio-pharmaceutical material. 

Based on Appendix A using LC-MS/MS (liquid chromatography with tandem mass spectrometry) and ELISA (Enzyme-linked Immunosorbent Assay), concentrations of cannabidivarinic acid (CBDVA), saponin, and ferulic acid were approximately 0.016 µg/mg, 1320 ng/µL, and 1.82 ng/µL, respectively, in hemp seed extract (HSE). In contrast, their concentrations in GHSEs were approximately 0.021 μg/mg, 1769 ng/µL, and 3.82 ng/µL, respectively (Appendix A). According to recent reports [38,39,40], saponin and ferulic acid, two major and minor compositions in hemp seeds, are upregulated in the germinated seeds of various plants. However, there is no report on increasing the concentration of CBDVA. Moreover, although these molecules are associated with preventing alopecia [41,42], there is no research on the effects of GHSE on alopecia. Thus, there is a possibility that altering these compounds affects the regulation of the markers, as shown in Figure 7. 

Consequently, although HSO prevented androgenic alopecia (Figure 7), E40 and GE2000 effectively prevented DT stress and activated mechanisms for hair formation through biological activities in three categories. 

## 4. Materials and Methods

### 4.1. Callus Induction and Purification of Exosomes, and Extraction of Germinated Hemp Seeds

After immersing 5 g of sterilized hemp seeds (Cheongsam) (88 company, Andong, Republic of Korea) in 1% H_2_O_2_ (hydrogen peroxide) for 24 h, the hydrated seeds were germinated for 3 days at 24 °C on Murashige and Skoog (MS; KisanBio, Seoul, Republic of Korea) germination media containing gibberellic acid (GA3, KisanBio), indole-3-acetic acid (IAA; KisanBio), and sucrose (KisanBio). Meristematic tissues were isolated from the germinated hemp seeds and cultured on MS (KisanBio) callus-inducing media containing 6-benzylaminopurine (KisanBio), IAA, and sucrose. After culturing the calli for 5 weeks on a callus transfer medium, exosomes were isolated and purified from the calli using the exoEasy Maxi Kit (DE; QIAGEN, Hilden, Germany), the CD68 Exo-Flow Capture Kit (SBI, Mountainview, CA, USA), and the exosome standards kit (STL; Sigma-Aldrich, St. Louis, MO, USA). GHSEs were prepared by treating the germinated seeds with 50% ethyl alcohol, followed by 12 h of incubation under 50% ethyl alcohol and hydrothermal treatment and 12 h of incubation under DW at 70 °C (hydrothermal extract; HE). Additionally, alterations in saponin and ferulic acid levels in the GHSEs and hemp seed extracts due to treatment with 50% ethyl alcohol were evaluated using ELISA kits (saponin; biorbyt, CA, UK, and ferulic acid; Creative Diagnostics, NY, USA) (Appendix A). 

### 4.2. Establishment of Treatment Dosage in Human Follicle Dermal Papilla Stem Cells

Proliferated HFDPSCs (Sku: M36007-08S; Celprogen, Torrance, CA, USA) were cultured on specific growth media (SKU: 36007-08; Celprogen) for 1 day at 37 °C and 5% CO_2_ under various treatment conditions (HE, AE30 to 70, and exosome extracts) to establish the treatment concentrations. The total RNA of the exposed cells was isolated using RiboEx reagent (GeneAll, Seoul, Republic of Korea), and cDNA was synthesized from the isolated RNA using Maxime RT PreMix (iNtRON, Seongnam, Republic of Korea) to evaluate and compare the effects between hydrothermal and alcoholic extraction. cDNA was amplified using primers (Table 1) and the following cycling parameters: 1 min at 95 °C, followed by 35 cycles of 35 s at 59 °C, and 1 min at 72 °C. The amplified DNA was estimated using iBright FL1000 and iBright Analysis Software 4.0.0 (Invitrogen, Waltham, MA, USA). 

### 4.3. Evaluating Metabolic Modulators of Alopecia

After exposure to various treatment conditions (Con, DT, GE2000, DT+GE2000, E40, E40+DT, HSO 1000 µg/mL) for 24 h, the total RNA in HFDPSCs (Sku: M36007-08S; Celprogen) was isolated from cells using RiboEx reagent (GeneAll; Republic of Korea). cDNA was synthesized from the isolated RNA using Maxime RT PreMix (iNtRON; Republic of Korea) and amplified with primers (Table 1) under the following cycling parameters: 1 min at 95 °C, followed by 35 cycles of 35 s at 59 °C, and 1 min at 72 °C. The amplified DNA was estimated using iBright FL1000 and iBright Analysis Software 4.0.0 (Invitrogen) and Prism 7 (GraphPad, Boston, MA, USA). HFDPSCs were exposed to various concentrations of GE, exosomes, and HSO (Cheongsam; 88 company) extracted under a low temperature (70 °C) and pressure to establish effective concentrations. β-catenin expression was evaluated at these concentrations using FITC-anti-β-catenin (Abcam, Cambridge, UK). The stained cells were analyzed using a flow cytometer (BD FACScalibur; BD Biosciences, San Jose, CA, USA), FlowJo 10.6.1 (BD Bioscience), and Prism 7 (GraphPad). 

### 4.4. Evaluating the Differentiating Patterns of HFDPSCs under the Two Materials

After exposure to various treatment conditions (Con, DT, GE2000, DT+GE2000, E40, and E40+DT) for 24 h, the total RNA in HFDPSCs (Sku: M36007-08S; Celprogen) was isolated from cells using RiboEx reagent (GeneAll). cDNA was synthesized from the isolated RNA using Maxime RT PreMix (iNtRON) and amplified (Table 1) under the following cycling parameters: 1 min at 95 °C, followed by 35 cycles of 35 s at 59 °C, and 1 min at 72 °C. The amplified DNA was estimated using iBright FL1000 and iBright Analysis Software 4.0.0 (Invitrogen). Cultured cells were fixed with 2% paraformaldehyde for 4 h and treated with 0.02% Tween 20 for 5 min. The treated cells were incubated with fluorescence-conjugated antibodies (three immunoglobulins), FITC-anti-K15 (Novus Biologicals, Centennial, CO, USA), and APC-anti-Sox2 (Abcam) at 37 °C for 2 days and evaluated using a flow cytometer (BD FACScalibur), FlowJo 10.6.1 (BD Biosciences), and Prism 7 (GraphPad).

### 4.5. Images of Differentiated HFDPCs and Germ Cells

The cultured cells were exposed to various treatment conditions (Con, DT, GE2000, DT+GE2000, E40, E40+DT, and HSO 1000 μg/mL), fixed with 2% paraformaldehyde for 4 h, and treated with 0.02% Tween 20 for 5 min. The treated cells were incubated with fluorescence-conjugated antibodies (three immunoglobulins), FITC-anti-K15 (Novus Biologicals), and APC-anti-Sox2 at 37 °C for 2 days. The stained cells were evaluated using a flow cytometer (BD FACScalibur), FlowJo 10.6.1 (BD Biosciences), and Prism 7 (GraphPad). 

### 4.6. Modulation of CD8^+^ T Cells by the Two Materials

The proliferated T cells (MOLT-4; ATCC, Manassas, VA, USA) were exposed to conditioned media (Con, GE2000CM, DTCM, DT+GE2000CM, E40CM, and E40+DTCM), fixed with 2% paraformaldehyde for 4 h, and treated with 0.02% Tween 20 for 5 min to evaluate the downregulation of inactivating factors. The treated cells were incubated with fluorescence-conjugated antibodies (three immunoglobulins), FITC-anti-CD8 (BD Biosciences), PE-anti-IL-2Rβ (BioLegend, San Diego, CA, USA), PerCP-anti-JAK1 (ThermoFisher, Waltham, MA, USA), and APC-anti-STAT1 (Abcam) at 37 °C for 2 days. Using population gating (CD8^+^ populations), the stained cells were evaluated with a flow cytometer (BD FACScalibur), FlowJo 10.6.1 (BD Biosciences), a fluorescence microscope (Eclipse Ts-2; Nikon, Shinagawa, Japan), imaging and cell count software (NIS-elements V5.11; Nikon), and Prism 7 (GraphPad). 

### 4.7. Concentration of IFNγ in CD8^+^ T Cells by the Two Materials

The proliferated T cells (MOLT-4; ATCC) were exposed to conditioned media (Con, GE2000CM, DTCM, DT+GE2000CM, E40CM, and E40+DTCM). Subsequently, the secreted IFNγ from stimulated CD8^+^ T cells was evaluated using IFNγ ELISA kits (ThermoFisher), AMR-100 (ALLSHENG, Hangzhou, China), and Prism 7 (GraphPad). 

## 5. Conclusions

In this study, E40 and GE2000 revealed potential functions against alopecia, including genetic modulation in HFDPSCs, the activation of cellular differentiation, and the modulation of the immune system. However, since this study was conducted at the cellular level, a limitation is that the in vivo effects remain unclear. Furthermore, the challenge of exosome productivity presents a hurdle for their application in the pharmaceutical market. To address these issues, we are currently investigating their effects using chick embryos, which exhibit early-stage feather follicle development. To overcome the limitations of exosome production, profiling microRNAs in callus exosomes will be crucial for the development of liposomal drugs. Positive evaluations of these experiments in the future will be pivotal in advancing the development of new therapies for androgenic alopecia. Moreover, GE2000 possesses the advantage of being applicable to the development of various products, potentially serving as a bridgehead for the advancement of the hemp industry. This suggests that it could play a crucial role in revitalizing the future natural materials market.

## Figures and Tables

**Figure 1 ijms-25-07823-f001:**
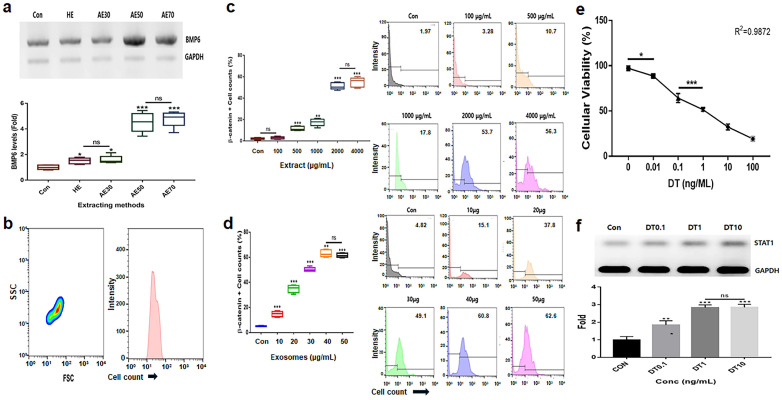
Establishment of exposure concentrations for biomaterials and stressors. (**a**) Comparison of hydrothermal extracts (HEs) and alcoholic extracts (AEs; AE%) for effective upregulation of marker levels. (**b**) Purification of induced exosomes derived from calli of germinated hemp seeds. (**c**–**f**) Establishment of exposure concentrations with germinated extracts (**c**), induced exosome (**d**), and dihydrotestosterone (DT; **e**,**f**) using evaluations of effective doses. ns, not significant; (* *p* < 0.05; ** *p* < 0.01; *** *p* < 0.001).

**Figure 2 ijms-25-07823-f002:**
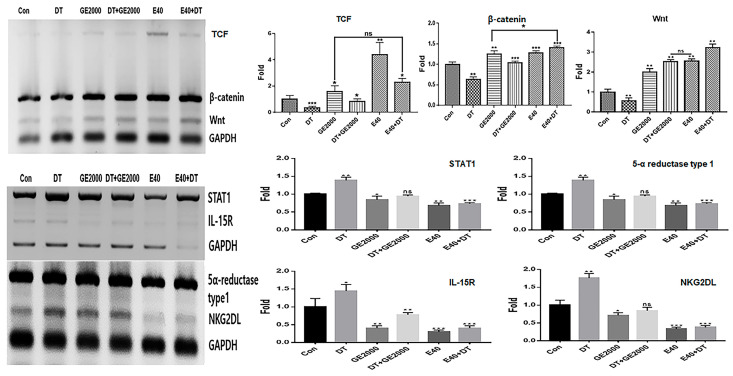
The levels of alopecia-modulating markers in hair dermal papilla stem cells under various conditions. These results show the levels of alopecia markers in hair follicle dermal papilla stem cells (HFDPSCs) under the germinated hemp seed extract (GHSE; 2000 µg/mL, GE2000), dihydrotestosterone (DT) 1 ng/mL, induced exosomes 40 μg/mL, DT1 after exposure to GE2000 (DT+GE2000), and DT1 after exposure to E40 (E40+DT). ns, not significant; (* *p* < 0.05; ** *p* < 0.01; *** *p* < 0.001).

**Figure 3 ijms-25-07823-f003:**
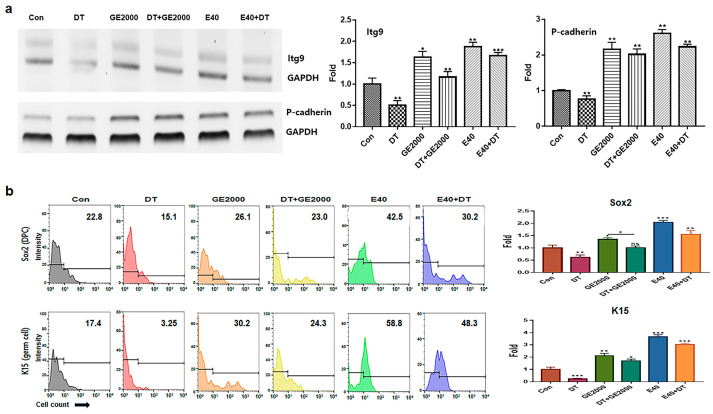
Differentiation of hair dermal papilla stem cells under various conditions. (**a**) Levels of markers for Itg9 (hair follicle dermal papilla cells) and P-cadherin (hair germ cells) genes under various conditions (Con, control; DT, dihydrotestosterone; GE2000, germinated hemp seed extract 2000 µg/mL; DT+GE2000, DT 1 ng/mL after exposure to GE2000; E40, induced exosomes 40 μg/mL derived from calli of germinated hemp seeds; E40+DT, DT1 after exposure to E40). (**b**) Expression of markers for Sox2 and K15 proteins under various conditions. ns: not significant; (* *p* < 0.05; ** *p* < 0.01; *** *p* < 0.001).

**Figure 4 ijms-25-07823-f004:**
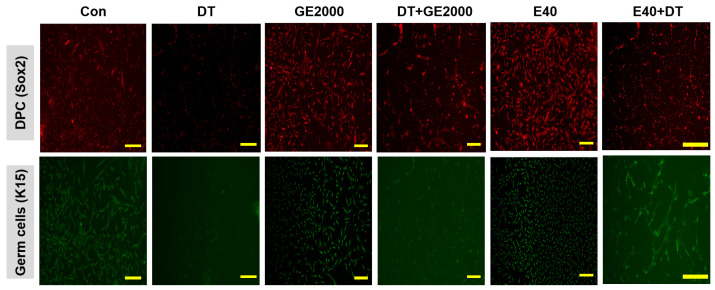
Images of the differentiated cells from hair follicle dermal papilla stem cells under various conditions. The results for the differentiated cells from hair follicle dermal papilla stem cells under different conditions (Con, control; DT, dihydrotestosterone; GE2000, germinated hemp seed extract 2000 µg/mL; DT+GE2000, DT 1 ng/mL after exposure to GE2000; E40, induced exosomes 40 μg/mL derived from the calli of germinated hemp seeds; E40+DT, DT1 after exposure to E40) (the scale bars = 30 μm).

**Figure 5 ijms-25-07823-f005:**
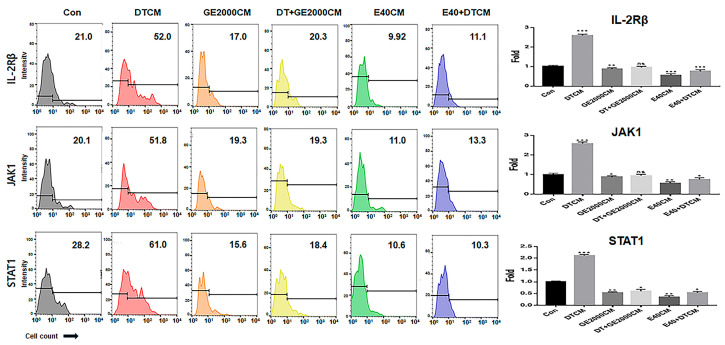
The expression of alopecia-activating markers in T cells under various conditioned media. The results for the expression of alopecia activators in CD8^+^ T cells exposed to various supernatants from HFDPSCs under different conditions (Con, control; DT, dihydrotestosterone; GE2000, germinated hemp seed extract 2000 µg/mL; DT+GE2000, DT 1 ng/mL after exposure to GE2000; E40, induced exosomes 40 μg/mL derived from the calli of germinated hemp seeds; E40+DT, DT1 after exposure to E40). These histograms are derived from the CD8^+^ population flow cytometry results. CM, conditioned medium; ns, not significant; (* *p* < 0.05; ** *p* < 0.01; *** *p* < 0.001).

**Figure 6 ijms-25-07823-f006:**
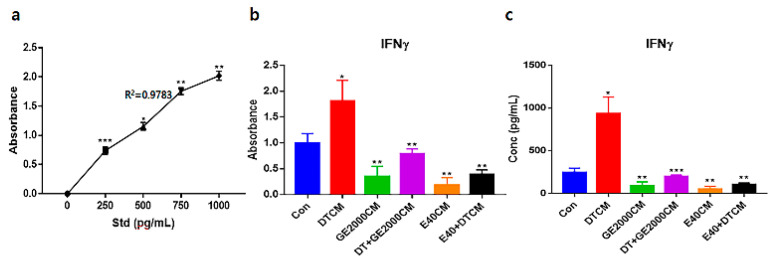
Evaluation of secreted interferon gamma from CD8^+^ cells exposed to various conditioned media. Standard curves (**a**) and evaluated absorbances and concentrations for interferon gamma (IFNγ) from hair follicle dermal papilla stem cells (HFDPSCs) exposed to various conditioned media (**b**,**c**). Evaluating concentrations of IFNγs based on (**a**,**b**) panels. R, correlation coefficient (* *p* < 0.05; ** *p* < 0.01; *** *p* < 0.001).

**Figure 7 ijms-25-07823-f007:**
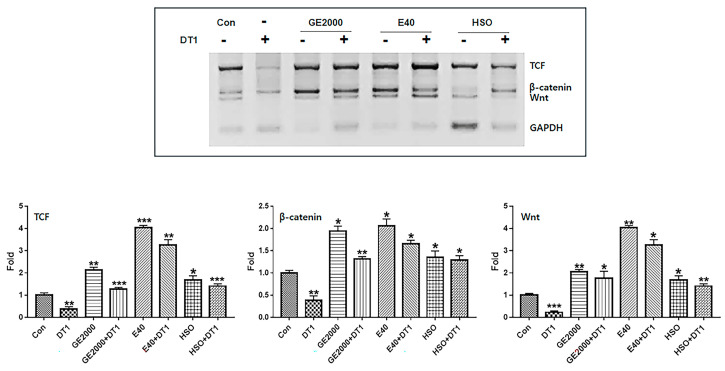
Comparison of two materials and hemp seed oil for preventive effects against alopecia. Levels of alopecia-preventing genes in hair follicle dermal papilla stem cells (HFDPSCs) exposed to three materials (GE2000, germinated hemp seed extract 2000 μg/mL; E40, induced exosomes 40 μg/mL from HFDPSCs exposed to 50% ethanolic extract; HSO, hemp seed oil 1000 μg/mL) under dihydrotestosterone 1 ng/mL (DT1) (* *p* < 0.05; ** *p* < 0.01; *** *p* < 0.001).

**Table 1 ijms-25-07823-t001:** Sequences for PCR primers.

Num	Target Gene	Sequence
1	*Sox2*	F: ATGGACAGTTACGCGCACA
		R: ATGCTGATCATGTCCCGGAG
2	*Itga9*	F: AAGGCCTGTCATTACGGTGG
		R: GCATCTGGCCCTTCTCCTTT
3	*STAT1*	F: TGTTATGGGACCGCACCTTC
		R: TTGGAGATCACCACAACGGG
4	*5α-* *reductase type 1*	F: GGCTGAGACCGGTGCG
		R: GGCTGAGACCGGTGCG
5	*P-cadherin*	F: GGGAGGCTGAAGTGACCTTG
		R: TGGAGCAACCACCCAATCTC
6	*Wnt*	F: TGCCTCACAAGGCATCAGTT
		R: GGAACCCATCAGGACACTCG
7	*TCF*	F: GGATGGCTATCCAGACGCTC
		R: AGTTTGGCACTCCAACTTCG
8	*β-* *catenin*	F: GAGCGGACCTACGGACTGTG
		R: CAGCGAGGTCACTTCCAACA
9	*NKG2DL*	F: CCTCTGTCACGGCTTTCCTT
		R: AATCCTTCCCTTGAGCACCG
10	*IL-15R*	F: CTCGGGTTTCAAGCGGAAGG
		R: TCGCGGATGCACTTGAGC
11	*GAPDH*	F: GTGGTCTCCTCTGACTTCAACA
		R: CTCTTCCTCTTGTGCTCTTGCT
	*GAPDH*	F: GGATTTGGTCGTATTGGGCG
		R: TCCCGTTCTCAGCCATGTAG

## Data Availability

The original contributions presented in this study are available on request from the corresponding author.

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
