# Peer review of "Effects of Biomaterials Derived from Germinated Hemp Seeds on Stressed Hair Stem Cells and Immune Cells"

_ijms, 2024, doi:10.3390/ijms25147823_

Round 1
Reviewer 1 Report
Comments and Suggestions for Authors
Dear Authors,
In general, the topic of utilizing germinated hemp seed extracts and exosomes for the prevention of androgenetic alopecia can be considered important from the perspective of enhancing both the physiological and psychological well-being of patients through innovative biochemical approaches.
While reading the research article, I encountered a few ambiguities, as mentioned below.
Results
Figures 1, 2, 3, and 5: It would probably be better divided into several individual figures. Even after a large zoom, the graphs cannot be clearly distinguished. Maybe a better resolution would resolve this.
Lines 100, 103, 137, 138, 141, and 145: Abbreviations should be explained when first used.
Conclusions
Lines 330-335: Expand the conclusion section with limitations of the study and future research recommendations.
Author Response
First, we appreciate for your comments and had English editing service again for this manuscript.
We revised our manuscript based on your comments and marked the revised sentences with underlines, a red color. The revised sentences based on common comments from reviewers were underlined in blue color in the file, marked version.
We revised the figures (Figures 1, 2, 3, and 5) and conclusion section and, added the full names for those (Lines 100, 103, 137, 138, 141, and 145).
Reviewer 2 Report
Comments and Suggestions for Authors
The manuscript entitled Effects of biomaterials derived from germinated hemp seeds for stressed hair stem cells and immune cells presented a detailed research aimed at evaluating the effect of germinated hemp seeds on alopecia. The paper is well written and well organized; nevertheless there are some revisions I recommend before its publication:
The introduction section is poorly written and it should better contextualize the phenomen at national level. Also reporting other similar studies investigating the research area. In these regards, could suggest an additional paragraph about study’s background. In addition, the research question and the aim of the study should be better explained.
Results are well discussed as well as discussion section. For this latter I could suggest to better explain literature main findings thus highlighting the novelty of your research. Conclusions are the weekest part of the manuscript. It should be totally rewritten highlighting the findings of your study, the limitations and future perspectives.
Comments on the Quality of English Language
Moderate english check required
Author Response
First, we appreciate for your comments and had English editing service again for this manuscript. We revised our manuscript based on your comments and marked the revised sentences with underlines, an orange color. The revised sentences based on common comments from reviewers were underlined in blue color in the file, marked version.
We revised the introduction and conclusion sections based on your comments.
Reviewer 3 Report
Comments and Suggestions for Authors
My main comment on this manuscript is that understanding of the results of the study is heavily dependent upon reference to the figures presented throughout the Results section. All the figures are complex but of low quality for determination of what they show because the detail is so small it is mostly unreadable even when magnified because the text elements become pixelated. Some of the description in the legends could be improved because it is complex and the description of the a), b), and c) elements, where these are present run so much into each other that it is difficult to read from one to another, especially as the text of the Results section largely refers to "results in the figures" which is incorrect because the figures are simply graphic representations of the results intended to clarify the text, so more textual description is required.
Figure 4 is almost completely obscured because there is such low contrast between the cells and the background and it would be better to narrow the field to a smaller number of cells showing the relevant colour change, or lack of it, but with better definition to distinguish them from the background coloured field. Also the chromatogram in Figure 7 requires more contrast.
There are several omitted words and confusions in the Discussion, the most obvious examples are:
Line 177: "restricted by cannabis regulation in many countries, in recent, many countries have eased or abolished the regulations" some wording is missing after recent.
Line 240: "These results suggest that the results of Figure 7 were correspondently with these analytic results." This sentence not only has a word that does not exist "correspondently" but also does not make sense in other ways and requires clarification.
Comments on the Quality of English LanguageGenerally the quality of English is good but there are sections that need some improvement such as parts of the Discussion, obvious examples of which are given in comments, and overall there is too much brevity of explanation, which is not incorrect in terms of language but does not demonstrate clearly what you think and would be improved by more extended comment
Author Response
First, we appreciate for your comments and had English editing service again for this manuscript. We revised our manuscript based on your comments and marked the revised sentences with underlines, a yellow color. The revised sentences based on common comments from reviewers were underlined in blue color in the file, marked version.
We revised the discussion, conclusion and backgrounds of fig4 and 7 based on your comments. Additionally, we revised all figures with clarification.
Reviewer 4 Report
Comments and Suggestions for Authors
REVIEW
Article:,, Effects of biomaterials derived from germinated hemp seeds for stressed hair stem cells and immune cells”
In the publication, the author discusses the important topic of androgenetic alopecia.
Androgenic alopecia affects the psyche of men and women. Undertaking research is justified and valuable.
Recommendations:
-please specify the purpose in the abstract (This study aims to demonstrate functions of GHSE and exosomes derived from calli of germinated hemp seeds for the prevention of alopecia) (stem cells of dermal hair papillae) (human hair follicle dermal papilla stem cells)
-The abstract does not provide information on what material the research was conducted on. Please complete
-Modulation of alopecia markers in dermal papilla stem cells of hair follicles using two materials (which??? Complete) Line 91
-Please correct the fragment below, it is unclear to the average reader:
,,Although this research suggested effective two biomaterial derived from hemp to pre-vent androgenic alopecia, effects of two materials in vivo is unclear. However, we have been researching the effects of two materials with chick embryo. The animal replace- ment experiment is required to apply as a cosmetic functional material. In the future, pos- itive evaluation of this experiment will be the cornerstone of the development of new drugs to therapy androgenic alopecia”
-The publication may include a drawing of the individual phases of hair growth with dihydrotestosterone DT receptors marked, so that the publication expands the group of readers.
After correction, please send the article to me again for verification.
Publication to be published, after minor corrections
Best Regards,

Author Response
First, we appreciate for your comments and had English editing service again for this manuscript. We revised our manuscript based on your comments and marked the revised sentences with underlines, a green color. The revised sentences based on common comments from reviewers were underlined in blue color in the file, marked version.
We revised the abstract section, the line 91 and the conclusion section with based on your comments. Additionally, we revised graphical abstract with hair growth cycles.
Round 2
Reviewer 2 Report
Comments and Suggestions for Authors
The authors incorporated reviewers suggestions and implemented the manuscript. Therefore, I would suggest minor comments to improve the manuscript, extending the conclusion section, because it is still the weakest part of the manuscript. It should concisly resume main findings, highlighting the limitations and future perspectives.
Comments on the Quality of English LanguageMinor English language editing required
Author Response
Thank you for your comments.
We re-revised the conclusion section based on your comment
Reviewer 3 Report
Comments and Suggestions for Authors
Thank you for making changes, you have largely addressed the comments I made and I suspect some of my comments about figures could not be further acted upon due to the typesetting process not allowing for large images. Although images are much improved for sharpness it is still necessary to increase the magnification several times in order to see clearly what some of the mini graphs in #Figures 1,3, and 5 demonstrate. Perhaps larger versions could be included as a Supplementary file.
Author Response
Thank you for your comments.
We improved the figures and provided them as supplementary data (S2)